# Accurate Prognosis Prediction of Pancreatic Ductal Adenocarcinoma Using Integrated Clinico-Genomic Data of Endoscopic Ultrasound-Guided Fine Needle Biopsy

**DOI:** 10.3390/cancers13112791

**Published:** 2021-06-03

**Authors:** Joo Kyung Park, Hyemin Kim, Dae-Soon Son, Nayoung K. D. Kim, Young Kyung Sung, Minseob Cho, Chung Lee, Dong Hyo Noh, Se-Hoon Lee, Kyu Taek Lee, Jong Kyun Lee, Kee-Taek Jang, Woong-Yang Park, Kwang Hyuck Lee

**Affiliations:** 1Department of Medicine, Samsung Medical Center, Sungkyunkwan University School of Medicine, Seoul 06351, Korea; mdsophie@gmail.com (J.K.P.); kimhyemin1120@gmail.com (H.K.); youngkyung.sung@gmail.com (Y.K.S.); ktcool.lee@samsung.com (K.T.L.); jongk.lee@samsung.com (J.K.L.); 2Department of Health Sciences and Technology, SAIHST, Sungkyunkwan University, Seoul 06351, Korea; 3Medical Research Institute, Sungkyunkwan University School of Medicine, Seoul 06351, Korea; 4School of Big Data Science, Data Science Convergence Research Center, Hallym University, Chuncheon 24252, Korea; biostat@hallym.ac.kr (D.-S.S.); chominseob1@naver.com (M.C.); 5Geninus Inc., Seoul 05836, Korea; nayoung.kim@kr-geninus.com (N.K.D.K.); chung.lee@kr-geninus.com (C.L.); 6Department of Internal Medicine, Eulji University Hospital, Daejeon 34584, Korea; laborhyo@naver.com; 7Division of Hematology-Oncology, Department of Internal Medicine, Samsung Medical Center, Sungkyunkwan University School of Medicine, Seoul 06351, Korea; sehoon.lee119@gmail.com; 8Department of Pathology, Samsung Medical Center, Sungkyunkwan University School of Medicine, Seoul 06351, Korea; kt12.jang@samsung.com; 9Samsung Genome Institute, Samsung Medical Center, Seoul 06351, Korea; 10Department of Molecular Cell Biology, Sungkyunkwan University School of Medicine, Suwon 16419, Korea

**Keywords:** targeted deep sequencing, pancreatic ductal adenocarcinoma, prognosis prediction, endoscopic ultrasound-guided fine needle core biopsy, clinico-genomic model

## Abstract

**Simple Summary:**

Around 10–20% of patients with pancreatic adenocarcinoma (PDAC) have a curative surgery, but most of them perform endoscopic ultrasound-guided fine-needle biopsy (EUS-FNB) for the pathologic confirmation. The aim of our retrospective study was to evaluate the feasibility of targeted sequencing using EUS-FNB minimal specimens for the prognosis prediction of PDACs. We found some clinical factors and genetic alterations significantly related to the metastasis and overall survival, and established a clinico-genomic model by using both clinical parameters and genetic alterations to predict the prognosis of patients with PDACs.

**Abstract:**

The aim of this study was to investigate the clinical utility of minimal specimens acquired from endoscopic ultrasound-guided fine-needle biopsy (EUS-FNB) and perform targeted deep sequencing as a prognosis prediction tool for pancreatic ductal adenocarcinoma (PDAC). A total of 116 specimens with pathologically confirmed PDAC via EUS-FNB were tested using CancerSCAN^®^ panel for a customized targeted deep sequencing. Clinical prognostic factors significantly associated with survival in PDACs were as follows: stage, tumor mass size, tumor location, metastasis, chemotherapy, and initial CA19-9 level. A total of 114 patients (98.3%) had at least a single genetic alteration, and no mutations were detected in two patients, although they were qualified for the targeted deep sequencing. The frequencies of major gene mutations responsible for PDACs were *KRAS* 90%, *CDKN2A* 31%, *TP53* 77%, and *SMAD4* 29%. A somatic point mutation of *NF1*, copy number alteration of *SMAD4*, and loss-of-function of *CDKN2A* were significantly associated genetic factors for overall survival. Moreover, *BRCA2* point mutation was related to liver metastasis. Finally, a clinico-genomic model was developed to estimate the prognosis of patients with PDAC based on clinical parameters and genetic alterations affecting survival in patients; 20 single nucleotide variants and three copy number variations were selected. Targeted deep sequencing on minimal specimens of PDACs was performed, and it was applied to establish a clinico-genomic model for prognosis prediction.

## 1. Introduction

Pancreatic ductal adenocarcinoma (PDAC) is the fourth leading cause of cancer-related mortality in the United States of America [1], and the fifth leading cause in Korea [2]. The five year overall survival (OS) rate for PDAC is approximately 7%, which is attributed to the typically advanced stage at diagnosis, combined with the lack of effective systemic chemotherapy available [3]. Predicting the prognosis and identifying patients who could benefit from specific therapies could improve the OS of patients with pancreatic cancer.

The factors responsible for the survival of patients with PDAC are poorly understood. The favorable prognostic factors of PDAC are primarily clinical factors, such as early disease stage, negative resection margins, and negative lymph nodes [4]. Current clinical prognostic factors produce limitations in predicting the prognosis of PDAC and selecting suitable patients for specific systemic chemotherapy regimens.

Most PDACs are unresectable when first diagnosed and endoscopic ultrasound-guided fine-needle biopsy (EUS-FNB) is the first choice for pathologic confirmation, and those minimal specimens can be widely used for research purposes compared to surgical specimens. Therefore, EUS-FNB has become a most important tool to explore the majority of PDAC tumor biology. Understanding the underlying molecular mechanisms of PDAC could promote the development of a new methodology for early diagnosis, and facilitate improvement in current approaches for PDAC treatment. Molecular profiling of tumor specimens has revealed potential targets for personalized therapy. Next-generation sequencing (NGS) allows parallel sequencing of DNA, which can advance the understanding of the underlying molecular pathophysiology of cancer [5]. Recent NGS genomic analyses of PDAC samples have revealed a complex mutational landscape from surgical specimens [6,7]. Minimal specimens of PDAC may be relatively hypocellular, meaning that small numbers of cancer cells can be outnumbered by stroma or inflammatory cells. Therefore, EUS-FNB specimens frequently yield an insufficient amount of DNA unfit for molecular analysis [8]. Some studies have described the feasibility of NGS using EUS-FNB acquired pancreatic tumor specimens, and demonstrated the superiority of NGS over conventional sequencing in detecting specific genomic alterations in EUS-FNB [9]. The aims of this study were to evaluate the feasibility of targeted sequencing from EUS-FNB minimal specimens, and to determine if the genomic signature of primary PDACs could be correlated to clinical parameters associated with the prognosis of PDACs.

## 2. Results

### 2.1. Baseline Characteristics

A total of 166 EUS-FNB samples were collected, and 116 samples passed the quality control test of DNA analytes (Appendix A) to obtain their genomic profiles through targeted deep sequencing. The schematic flow of study design is shown in Figure 1. Survival data was obtained from the hospital database, and information on mortality was collected via telephone calls in cases of loss to follow-up. The clinical and pathologic characteristics of the 116 minimal specimens from the study patients are shown in Table 1. The median age was 64 years old (range 26–83 years) and male to female ratio was 67:49 (1.4:1). According to the 8th American Joint Committee on Cancer (AJCC) staging system, one (0.9%) patient was stage IB, six (5.2%) were stage IIA, eight (6.9%) were stage IIB, 26 (22.4%) were stage III, and 75 (64.6%) were stage IV. The most common location of the tumor was the head (42.2%), followed by the body (35.4%) and tail (22.4%) of the pancreas. The most common metastatic site was the liver (60%). Among the 116 study patients, 13 (11.2%) patients underwent surgical resection with curative intention, and 11 (84.6%) of them followed by adjuvant therapy. A total of 92 (79.3%) patients were treated with palliative chemotherapy. In addition, 24 (20.7%) patients were treated with best supportive care only.

### 2.2. Clinical Prognostic Factors Affecting OS and Clinical Model

The clinical factors associated with the OS of patients with PDAC were evaluated. Median OS was 8.4 months (range 0.77–103.0 months). Age, gender, BMI, CA 19-9 level, chemotherapy, tumor mass size, tumor location (head, body, tail), clinical stage, and metastasis were used for analysis (Figure 2A–F). Univariate cox proportional-hazards analysis revealed the prognostic significance of stage (HR = 1.55 (95% CI, 1.22–1.97); *p* = 0.0001), metastasis (HR = 1.91 (95% CI, 1.27–2.88); *p* = 0.0015), tumor location (HR body = 1.36 (95% CI, 0.88–2.08); HR tail = 1.97 (95% CI, 1.21–3.21); *p* = 0.0273), tumor size (HR = 1.97 (95% CI, 1.34–2.90); *p* = 0.007), CA19-9 level (HR = 1.58 (95% CI, 1.08–2.32); *p* = 0.0183), and chemotherapy (HR = 0.33 (95% CI, 0.21–0.52); *p* < 0.001). However, age (HR = 0.94 (95% CI, 0.65–1.36); *p* = 0.7467), gender (HR = 1.08 (95% CI, 0.74–1.57); *p* = 0.7057), BMI (HR = 0.91 (95% CI, 0.63–1.32); *p* = 0.6142), T classification (HR = 1.05 (95% CI, 0.79–1.38); *p* = 0.7359), N classification (HR = 1.40 (95% CI, 0.93–2.09); *p* = 0.0985), and smoking status (HR ex-smoking = 0.98 (95% CI, 0.60–1.58); HR smoking = 1.76 (95% CI, 1.01–3.05); *p* = 0.1553) were not significantly related to OS (Appendix A). For the variables included in the optimized clinical model, chemotherapy, mass size, and CA19-9 were selected using a stepwise method, and age and stage were adjusted (Figure 3A). This model could separate the high-risk group (median OS, 8.33 months) from the low-risk group (median OS, 9.93 months) for PDAC (HR = 2.11 (95% CI, 1.44–3.10); *p* = 0.0001) (Figure 3B).

### 2.3. Characteristics of Genetic Alterations in EUS-FNB Samples

To investigate genetic alterations, targeted sequencing using the CancerSCAN^®^ panel was performed with EUS-FNB specimens of PDAC. Targeted deep sequencing (Mean Coverage 957x, Appendix A) identified somatic DNA alterations, including single nucleotide variants (SNVs), small detect insertions and deletions (InDels) and copy number variations (CNVs). Among 116 samples of PDAC patients, there were no mutations detected in two of the samples, and 114 samples (98.3%) showed variable mutations with different frequency (Figure 4A). Each patient had a different number of genetic alterations, ranging from 0 to 14 (median = 4). Significant recurrent mutations were identified in *KRAS*, *TP53*, *CDKN2A*, *ARID1A*, *ATM*, *STK11*, *ARID1B*, *NOTCH1*, and *BRCA2* (Figure 4A), which were also recurrently mutated in other PDAC tumors [10,11,12]. The mutation frequencies of major oncogenes and tumor suppressor genes responsible for PDACs [3] were: *KRAS* 90%, *TP53* 77%, *CDKN2A* 31%, and *SMAD4* 29% (Figure 4A). Among genetic alterations, SNVs, especially missense mutations were dominant, and C > T transition was the most frequently found (Appendix A). A percentage of variant allele fraction (VAF) was the highest in *ATM* (43%), followed by *BRCA2* (40%), *NOTCH1* (39%), *ARID1B* (24%), *CDKN2A* (24%), *TP53* (23%), *STK11* (21%), *ARID1A* (17%), *KRAS* (20%), and *SMAD4* (10%) (Figure 4B).

### 2.4. Genetic Alterations Associated with OS and Metastasis

Genetic alterations were analyzed to explore their association with OS. Alterations of *KRAS*, *CDKN2A*, *TP53*, and *SMAD4* genes were not significantly associated with OS. Univariate cox regression analysis indicated that PDAC patients with somatic mutations in *NF1* (3.35 vs. 9.63 months; HR = 6.21 (95% CI, 2.18–17.66); *p* = 0.0061) showed worse prognosis (Figure 5A), and CNV deletion of *SMAD4* (22.2 vs. 8.50 months; HR = 0.40 (95% CI, 0.15–1.08); *p* = 0.0367) was related to longer survival (Figure 5B). Moreover, loss-of-function (LOF) mutation of *CDKN2A* was associated with poor survival (HR = 1.93 (95% CI, 1.25–2.97); *p* = 0.0025, Figure 5C), which was validated with The Cancer Genome Atlas (TCGA) data (PMID:29625048) (HR = 1.71 (95% CI, 1.15–2.56); *p* = 0.0052, Figure 5D).

Overall, 75 patients presented with distant metastasis among a total of 116 patients with PDAC (75/116, 64.6%). This was analyzed to explore the association between genetic alteration and metastasis. Among 75 PDAC patients with distant metastasis, 45 patients (45/75, 60%) displayed liver metastasis, and 30 patients (30/75, 40%) had distant metastasis to other organs except the liver. Other metastatic sites included lung (9/30, 30%), peritoneum (20/30, 66.7%), lung (1/30, 3.3%), and distant LN (7/30, 23.3%). Mutations of *BRCA2* gene were significantly related to the metastasis sites (Table 2). The *BRCA2* mutation was associated with liver metastasis (*p* = 0.0162).

### 2.5. Development of Clinico-Genomic Model

To develop a prognostic model for PDAC patients, additional useful genetic alterations were included based on the clinical model. Since genetic testing provides additional information to the clinical model, the clinico-genomic model used the forward variable selection method as a way to find genes that additionally contribute to the clinical model (Figure 6A). Each gene was used as a binary variable according to the presence or absence of alteration, such as SNVs and small InDels in each gene region. When each gene has CNV, it was treated as a separate variable. It successfully estimated the high-risk group (median OS, 5.95 months) and the low-risk group (median OS, 15.27 months) based on clinical factors and genetic alterations (HR = 6.06 (95% CI, 3.84–9.55); *p* < 0.001, Figure 6B).

## 3. Discussion

In the present study, we analyzed various clinical factors and genetic alterations in relation to the survival of PDAC patients. Moreover, we developed a clinico-genomic model and successfully predicted the clinical outcome for PDACs.

Approximately 10 to 20% of patients with PDAC are operable [3], thus, personalized genetic information is not available for most patients. EUS-FNB is a primary method to acquire tissue samples of pancreatic cancer for pathologic confirmation. We conducted targeted sequencing with EUS-acquired tissue specimens, which are more accessible than surgical specimens. Success rates for NGS analysis of solid tumor tissue specimens were dependent on the types of samples being sequenced. The rate of NGS success was the highest in resection specimens (97%), reduced within biopsy specimens (80%), and was lowest for cytologic specimens (50%) [13]. Several studies have reported that the adequacy of EUS tissue acquisition samples for deep sequencing ranged from 60 to 100% in pancreatic solid mass [14]. In our study, we successfully performed targeted sequencing with EUS-FNB samples in 116 patients (70%) among 166 patients with PDAC. Interestingly, a report showed that the observed EUS-FNA cytology mutational spectrum detected with a commercial 160 cancer gene panel was broader than that of the corresponding surgical pathology specimens [15]. Advanced NGS technologies can be applied to the EUS sample, which allowed the unveiling of potential genomic biomarkers and therapeutic targets [14]. Several studies have reported the adequacy of EUS samples for NGS. Elhanafi et al. [16] analyzed 167 EUS samples of PDAC patients by targeted sequencing with a 47 gene panel, and frequency of genomic alterations were *KRAS* (88%), *TP53* (68%), and *SMAD4* (16%). Young et al. [17] performed NGS using a customized gene panel (287 genes) with formalin-fixed paraffin-embedded EUS samples, and the most common mutations were observed in *KRAS* (78%), *TP53* (74%), *CDKN2A/B* (35%), *SMAD4* (17%), and *PTEN* (13%). Among 116 samples, somatic mutations were identified in 114 samples (98.3%). The missing two samples without any detected mutations might have few neoplastic nuclei due to the minimal tissues of EUS-FNB. For main genetic drivers (*KRAS*, *TP53*, *CDKN2A*, and *SMAD4*) in PDAC carcinogenesis [3,6,7], *KRAS* was found to be the most commonly mutated gene (90%) in our PDAC specimens, and *TP53* (77%), *SMAD4* (29%), *CDKN2A* (31%), and *BRCA2* (4%) were also commonly mutated at comparable rates with other studies using EUS samples [14] and surgical samples [7] (Appendix A). Numerous other genes harbor mutations at lower frequencies. This indicates that we successfully performed targeted panel sequencing with minimal EUS-FNB specimens.

Known prognostic factors related to survival in patients with PDAC are clinicopathologic factors, including CA19-9 levels, disease stage, chemotherapy treatment, performance status, surgical margin status, perineural invasion, and blood vessel invasion [18]. We also identified tumor location and tumor mass size as independent prognostic factors (Figure 2).

Among the genes in the CancerSCAN^®^ panel, SNVs of *NF1* and deleted copy number loss of *SMAD4* were associated with survival in univariate analysis (Figure 5A,B). A report showed that *NF1* deficiency correlated with estrogen receptor phosphorylation and poor survival in breast cancer [19]. Although there is a case report showing an association of *NF1* with OS in pancreatic endocrine tumors [20], *NF1* mutation in PDAC has not been reported in terms of clinical outcome. *SMAD4* deletion has been commonly reported in 50% of PDAC, resulting in aberrant signaling by the transforming growth factor beta (TGF-β) cell surface receptor [3]. Patients with PDAC with SMAD4 protein expression had significantly longer survival [21]. Patients having *SMAD4* gene inactivation, by either intragenic mutation or homozygous deletion, had significantly worse survival than patients with intact *SMAD4* [22]. Recently, *SMAD4* and *NF1* mutations were reported as potential biomarkers for poor prognosis to cetuximab-based therapy in metastatic colorectal cancer patients [23]. Moreover, patients with LOF mutation of *CDKN2A* showed the poor outcome in Figure 5C, and it was also validated in the cohort data from TCGA (Figure 5D). It was reported that somatic mutations of *CDKN2A*, a tumor suppressor factor, are present in up to 95% of pancreatic tumors [24]. The inactivation of *CDKN2A* induced by homozygous deletion or promoter hyper-methylation and point mutation contributes to cancer progression in various cancers, like PDAC [25,26,27].

Integration of genomic and transcriptional features in pancreatic cancer revealed that cell cycle progression was increased in PDAC metastases with driver gene (*KRAS*, *TP53*, *CDKN2A*, *SMAD4*) loss [28]. Moreover, mutations in known driver genes in the metastases of individual patients with PDAC were remarkably uniform [29], and no significant differences were identified at the gene or pathway level when comparing genomic alterations between primary and metastatic pancreatic tumors [30]. Many studies have reported some genes which may be involved in tumor metastasis, such as *KAI1* and *nm23-H1* in pancreatic cancer [31]. *KAI1*, a putative metastasis suppressor gene, appears to be crucial for the development of distant organ metastasis, since its expression was downregulated in lymph node and distant metastasis [32]. Stratford et al. analyzed the gene expression profiles of primary tumors from patients with localized compared to metastatic disease, and identified a six-gene signature (*FOSB*, *KLF6*, *NFKBIZ*, *ATP4A*, *GSG1*, *SIGLEC11*) associated with metastatic disease [33]. These genes were prognostic for survival in PDAC. We also analyzed genetic alterations associated with metastasis, in which *BRCA2* mutation was found to be related to liver metastasis rather than other sites, such as lung and peritoneum (Table 2). *BRCA2* germline mutation is also one of the most common causes of familial pancreatic cancer. *BRCA2*-associated PDAC has a more favorable outcome than non-*BRCA*-associated PDAC [34]. Moreover, *BRCA2* mutations made the disease biologically more chemosensitive, which, consequently, prolong survival despite the prognostically unfavorable disease [35]. Patients carrying *BRCA2* pathogenic mutations are more likely to progress to metastasis in prostate cancer [36]. This is the first report that missense mutations in *BRCA2* were relevant in PDAC patients with liver metastasis instead of other sites. However, further functional studies are needed to elucidate these mechanisms.

By combining clinicopathologic factors with genetic alterations, we established an integrated clinico-genomic model to predict the survival of patients with PDAC (Figure 6). Five clinical factors, including age, stage, chemotherapy, tumor mass size, and CA19-9 level, and 23 genetic alterations were selected based on the targeted panel sequencing profile using a stepwise variable selection method. The clinico-genomic model was developed and successfully estimated the prognosis of PDAC. Clinical parameters such as age, stage, chemotherapy, tumor mass size, and plasma CA19-9 level could be collected soon after diagnosis, and EUS-FNB is performed for initial PDAC diagnosis. Therefore, it seems that the clinico-genomic model is useful for prognosis prediction of PDAC patients at the early period of diagnosis with genetic analysis using EUS-FNB samples. However, these require further investigation with more patients with PDAC.

The present study had several limitations. The study design is retrospective, and patient populations were heterogeneous. They received different treatment regimens, and were diagnosed at various stages (resectable and unresectable PDAC). Moreover, we excluded some patients who proceeded directly to surgical resection without EUS-FNB, and they might have had a better prognosis. We performed targeted sequencing using CancerSCAN^®^ panel instead of whole genome sequencing, which made it impossible to evaluate the correlation between genes not included in the panel and the survival of patients with PDAC. Moreover, we could not verify the clinico-genomic model with the open database, like TCGA, because it was not eligible for some clinical factors, such as chemotherapy, tumor mass size, and CA19-9 level used for this model. Only the validation of genetic alteration related to survival in pancreatic cancer was performed with the TCGA database (Figure 5). Finally, most novel genetic alterations correlated with survival, metastasis, metastatic sites, and chemotherapy responses and did not elucidate the mechanism of gene function in terms of related factors.

We demonstrated the applicability of targeted NGS and detection of novel genetic mutations associated with prognosis and metastasis in PDAC. Genetic information, in addition to currently utilized clinical information, could help to predict the prognosis of patients, which was proved by the establishment of a clinico-genomic model for prognosis at diagnosis. This provides a more comprehensive view of the genomic landscape of PDAC, with implications for improved clinical decision making in the context of the implementation of tailored therapies. Further functional investigations and validation studies are needed to fully elucidate the mechanism of novel genetic mutations.

## 4. Materials and Methods

### 4.1. Patients

Overall, 166 patients with pathologically confirmed PDAC who underwent EUS-FNB between 30 November 2011 and 31 December 2014 at the Samsung Medical Center were enrolled. Among them, 116 who passed a DNA quality control test were analyzed for their genomic profiles by targeted CancerSCAN^®^ panel sequencing. Written informed consent was obtained from each patient. This study was approved by the Institutional Review Board (IRB) of Samsung Medical Center (IRB No. 2014-04-048). Clinicopathologic data was collected, including age, gender, body mass index (BMI), smoking status, performance status (Eastern Cooperative Oncology Group, ECOG), operations, chemotherapy responses, tumor size, tumor location, overall/TNM stage of the disease, serum levels of carcinoembryonic antigen (CEA), carbohydrate antigen 19-9 (CA19-9), and overall survival. Electrical medical records were reviewed and clinical AJCC 8th system were used for the staging.

### 4.2. EUS-FNB Procedure

All EUS-FNB procedures were performed to obtain the primary pancreatic cancer tissue specimen under conscious sedation by experienced endosonographers with GF-UE160-AL linear EUS apparatus (Olympus, Tokyo, Japan) equipped with an Aloka ProSound SSD-5000 (Wallingford, CT, USA). EchoTip^®^ Ultra Endoscopic Ultrasound Needle or EndoTip ProCore^®^ HD Ultrasound Biopsy Needle (19, 22, or 25 gauges; Wilson–Cook Inc., Winston-Salem, NC, USA) were used, allowing several passes under direct endosonographic visualization to obtain sufficient cellular aspirates for cytologic analysis. A transgastric approach was performed on lesions in the body or tail of the pancreas, and a transduodenal approach was used for lesions in the head or uncinate process. For tissue retrieval, the stylet was introduced into the needle, or the needle was flushed with a 5 to 10 mL air-filled syringe. The extruded material was first used for primary gross inspection, and remaining tissues were snap-frozen in a screw vial and stored at −80 °C for DNA extraction. Informed consent for purpose and complication of EUS-FNB was obtained from all patients prior to procedure.

### 4.3. Targeted Deep Sequencing

Genomic DNA was extracted using a QIAamp DNA mini kit (Qiagen Inc., Valencia, CA, USA) according to the manufacturer’s instruction. DNA concentration and purity were checked using a Nanodrop 8000 UV-Vis spectrometer (Thermo Scientific, Waltham, MA, USA) and Qubit 2.0 Fluorometer (Life Technologies, Grand Island, NY, USA). The degree of DNA degradation was measured using a 200 TapeStation Instrument (Agilent Technologies, Santa Clara, CA, USA). Qualified DNA was sheared using a focused-ultrasonicator (Covaris, Woburn, MA, USA) and used for library construction using CancerSCAN^®^ (Cancer Somatic mutation call for Clinical reports with Annotation) version 1 based on a customized SureSelect XT reagent kit (Agilent, Santa Clara, CA, USA). CancerSCAN^®^ panel, a targeted sequencing platform designed at Samsung Medical Center, offered flexibility to include target genes curated from the literature as requested by the researchers and clinicians. These selected genes covered variants associated with the targeted cancer therapies (i) approved by the Korean MFDS and US FDA, (ii) in the clinical trials at the Precision Oncology Clinic at Samsung Medical Center, or (iii) reported as having association with response of therapy in the public databases and the literature. It was designed to enrich the exons of 83 genes (Appendix A), covering 366.2 kb and 2.59 Mb of the human genome [37]. After the enriched exome libraries were multiplexed, the libraries were sequenced on the HiSeq 2500 sequencing platform (Illumina, San Diego, CA, USA). Sequence reads were mapped to the human genome (hg19) using the Burrows–Wheeler Aligner. Duplicate read removal was conducted using Picard and SAMtools. Local alignment was optimized by The Genome Analysis Toolkit. Variant calling was only conducted in targeted regions of CancerSCAN^®^. To detect SNVs, results of three types of variant callers (MuTect, Vardict, and LoFreq) were integrated to increase sensitivity. Pindel was used to detect InDels. CNVs were calculated for targeted regions by dividing read depth per exon by estimating normal reads per exon using an in-house reference. In analysis pipeline, we first removed the very common germline variant based on the population frequency after mutation calling, and flagged the benign mutation based on the known public database, pathogenicity database, and in-house database (data from about 6000 people). TCGA (PMID: 29625048) data was downloaded from cBioPortal, and we compared binary gene mutation status, such as nonsynonymous, truncation, and copy number alteration with survival. R version 4.0.2 was used for statistical analyses, and survival v3.1.12 and survminer v0.4.8 packages used for survival analysis.

### 4.4. Clinical and Clinico-Genomic Model

We used overall survival time as the endpoint of survival analysis. The clinical model was constructed using only clinical variables, and both clinical and genetic variables were included to develop a clinico-genomic model. In order to explain the necessity of genetic testing in clinic, initially, an optimal clinical model was developed using only clinical parameters. To prove clinical usefulness of genetic testing, a clinico-genomic model should show better performance than a clinical model. Therefore, a clinico-genomic model was developed by adding genetic parameters that help to improve performance based on the clinical model (similar to the forward variable selection method). All statistical models were developed with multiple Cox’s proportional hazard regression method, which uses clinical parameters and mutation information for each gene as independent variables, and overall survival time and censoring data as response variables. The score calculated in this regression model was divided based on the median of calculated Cox score to separate the high-risk group and the low-risk group. Overall survival performance of each group was shown using a Kaplan–Meier graph, and log-rank method was used for significance testing.

### 4.5. Statistical Analyses

For all of the tests, significance level was set at 5% and reported as two-tailed p-values. Statistical analyses were carried out using R software for Linux version 3.6.2 (The R Foundation, St. Miami, FL, USA).

## 5. Conclusions

It was successful to perform targeted sequencing with minimal specimen from EUS-FNB, and analyze genetic alterations associated with the prognosis of patients with PDAC. Combined with clinical information, genetic alterations were useful to predict the survival of PDACs.

## Figures and Tables

**Figure 1 cancers-13-02791-f001:**
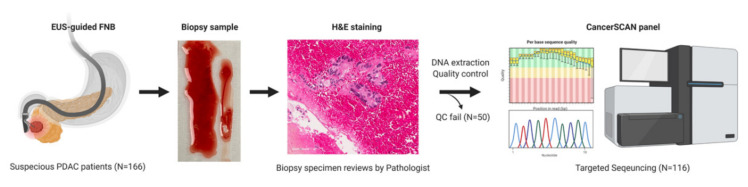
The schematic flow of study design. Patients suspected of PDAC (N = 166) performed endoscopic ultrasound-guided fine-needle biopsy (EUS-FNB) to obtain pancreas specimens, which were examined by pathologists after hematoxylin & eosin (H&E) staining. DNAs were extracted from each PDAC specimen, and DNA analytes (N = 116) passing a quality control (QC) test were sequenced by using CancerSCAN^®^ panel. PDAC, pancreatic ductal adenocarcinoma.

**Figure 2 cancers-13-02791-f002:**
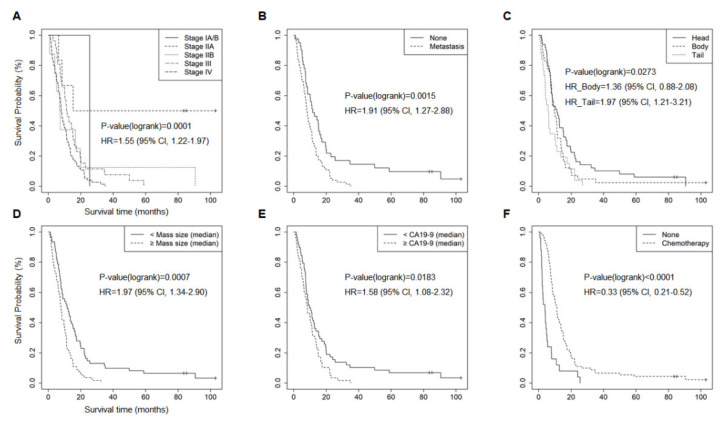
Kaplan–Meier plots for clinical factors significantly associated with survival. The Kaplan–Meier plots demonstrate six clinical factors with a significant (*p* < 0.05) association with the survival of patients with PDAC. (**A**) Stages, (**B**) metastasis, (**C**) tumor location, (**D**) tumor mass size, (**E**) CA19-9 level, and (**F**) chemotherapy. Hazard ratio (HR), confidence interval (CI), and *p*-value were obtained from univariate Cox proportional hazard test.

**Figure 3 cancers-13-02791-f003:**
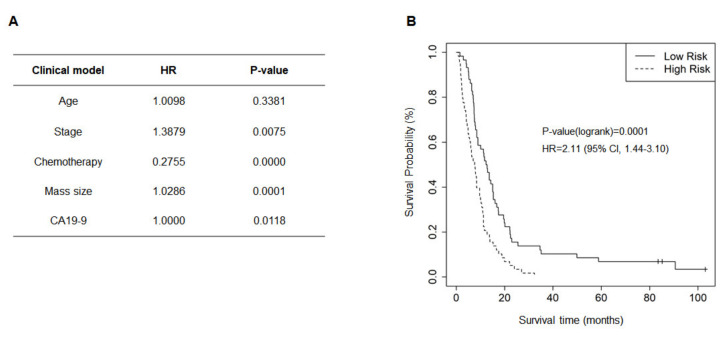
Clinical model. A clinical model considering various clinical parameters were developed to predict the survival of patients with pancreatic ductal adenocarcinoma (PDAC). (**A**) Selected clinical factors to establish the clinical model. (**B**) Kaplan–Meier plot for the high-risk group and the low-risk group of PDAC patients predicted by the clinical model (*p* = 0.0001).

**Figure 4 cancers-13-02791-f004:**
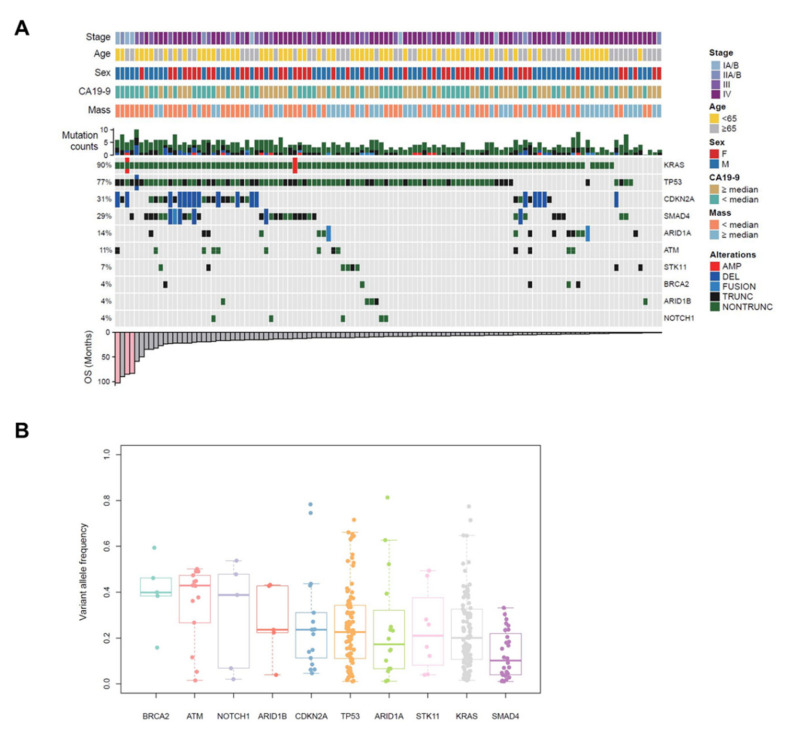
Landscape of genomic alterations identified by targeted deep sequencing in EUS-FNB specimens of patients with PDAC. (**A**) Co-mutation plot displaying integrated genomic data for 116 EUS-FNB specimens displayed as columns, including somatic mutations and somatic copy number variations for significantly mutated genes listed at the right (Top 10). The percentage of PDAC samples with an alteration of any type is noted at the left. Stage, age, sex, CA19-9 level, tumor mass size, and the number of any genetic alterations per each sample is presented on the top. Overall survival (OS) of each patient is shown on the bottom. (**B**) Variant allele frequency (VAF) distributions are analyzed, and median values are indicated by the central bar in the box and whiskers plots. PDAC, pancreatic ductal adenocarcinoma; EUS-FNB, endoscopic ultrasound-guided fine-needle biopsy; AMP, copy number amplification; DEL, copy number deletion; FUSION, structure variation; TRUNC, truncating mutation; NONTRUNC, non-truncating mutation.

**Figure 5 cancers-13-02791-f005:**
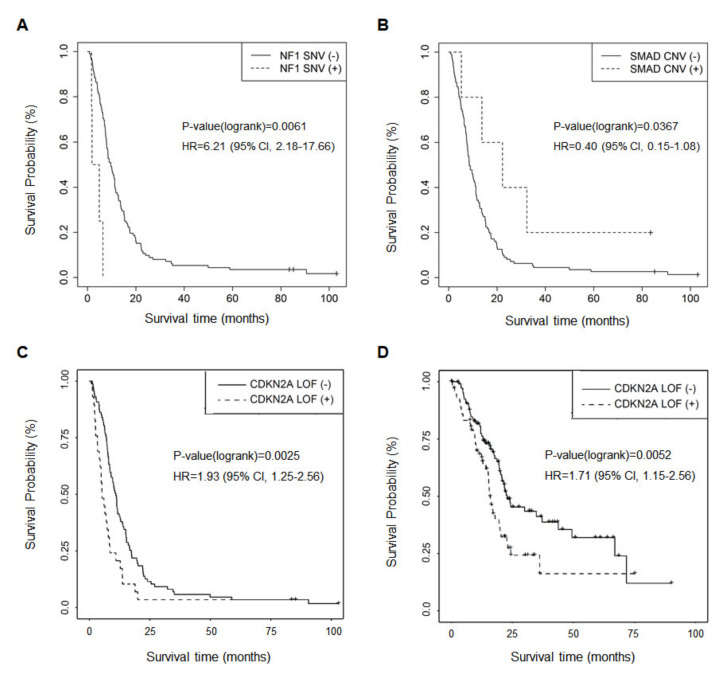
Kaplan–-Meier plots for genetic alterations significantly associated with survival. The Kaplan–Meier plots indicate genes significantly (*p* < 0.05) associated with the survival of patients with pancreatic ductal adenocarcinoma (PDAC). (**A**) Single nucleotide variant (SNV) of *NF1* (*p* = 0.0061), (**B**) copy number variation (CNV) of *SMAD4* (*p* = 0.0367), and (**C**) loss-of-function (LOF) mutation of *CDKN2A* (*p* = 0.0025) were related to survival in PDAC. (**D**) LOF mutation of *CDKN2A* and worse prognosis was verified by TCGA data.

**Figure 6 cancers-13-02791-f006:**
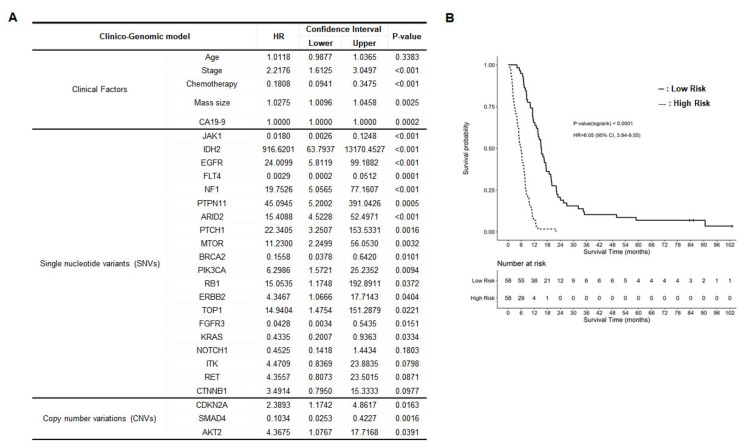
Clinico-genomic model. A clinico-genomic model considering clinical parameters and genomic alterations was developed to predict patient survival. (**A**) Selected factors to establish the clinico-genomic model. (**B**) Kaplan–Meier plot for the high-risk group and the low-risk group of patients with pancreatic ductal adenocarcinoma (PDAC) predicted by the clinico-genomic model (*p* < 0.0001).

**Table 1 cancers-13-02791-t001:** Clinicopathologic Characteristics of Patients with Pancreatic Cancer.

Variable	*n* = 116
Age, years (median ± SD)	64.0 ± 10.4
Gender	
Male	67 (57.8)
Female	49 (42.2)
BMI (median ± SD)	22.1 ± 3.1
Performance status (ECOG)	
0	76 (65.5)
1	36 (31.0)
2	3 (2.6)
3	1 (0.9)
CEA (ng/mL) (median ± SD)	2.8 ± 34.3 (13 missing)
CA19-9 (U/mL) (median ± SD)	393.0 ± 27295.3
Smoking, n (%)	
Never	78 (67.2)
Former	22 (19.0)
Current smoker	16 (13.8)
Treatment, n (%)	
Operation	
No	103 (88.8%)
operation with curative intention	13 (11.2%)
Chemotherapy	
No	24 (20.7%)
Yes	92 (79.3%)
Tumor mass size (cm) (median ± SD)	4.0 ± 1.6
Tumor location, n (%)	
Head	49 (42.2)
Body	41 (35.4)
Tail	26 (22.4)
Stage (AJCC 8th edition), n (%)	
IA/B	1 (0.9)
IIA	6 (5.2)
IIB	8 (6.9)
III	26 (22.4)
IV	75 (64.6)
Metastasis, n (%)	
None	41 (35.3)
Liver	45 (38.8)
Other sites except liver	30 (25.9)

SD, standard deviation; BMI, body mass index; ECOG, Eastern Cooperative Oncology Group; CEA, carcinoembryonic antigen; CA19-9, carbohydrate antigen 19-9; AJCC, The American Joint Committee on Cancer.

**Table 2 cancers-13-02791-t002:** Genetic Alterations Significantly Associated with Metastatic sites.

Mutaion	None (*n* = 41)	Liver (*n* = 45)	Other Sites (*n* = 30)	*p*-Value
BRCA2 mutation				0.0162
No	41	40	30	
Point mutation	0	5	0	

## Data Availability

The data presented in this study are available on request from corresponding authors.

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
