# Peer review of "Accurate Prognosis Prediction of Pancreatic Ductal Adenocarcinoma Using Integrated Clinico-Genomic Data of Endoscopic Ultrasound-Guided Fine Needle Biopsy"

_cancers, 2021, doi:10.3390/cancers13112791_

Round 1

Reviewer 1 Report

Reviewer’ Understanding and General Comment

The manuscript from Kim and colleagues describes the genetic alterations captured in a cohort of advanced pancreatic cancers (n = 116). The genetic analysis has been conducted using a high-coverage targeted sequencing panel that explore around 2.6 Mb of genomic space and target 83 cancer related genes. The information that can be retrieved from the panel is diverse and ranges from single-nucleotide variants to copy-number variations. The list of genes that can be found in the supplementary file suggests that some PDAC cancer genes (especially chromatin modifiers such as KDM6A and KMTs) are missing, but It does include core PDAC genes (KRAS, TP53, CDKN2A, SMAD4). The scope of this study is pretty clear, and it relates to the possibility of building a classifier that includes molecular and clinicopathological information to predict outcomes in this type of cohort. Given the paucity of molecular study conducted on advanced pancreatic cancer cohort, I think this is an interesting manuscript to the field. However, there are several concerns from this reviewer, and I would suggest the authors reducing the prominence of certain statements that are not fully supported by the evidences provided here.

Specific comments

  1. English needs revising and there are several typos in the text and the figures of the manuscript (e.g., top 10 “mutatied” genes in supplementary 2).
  2. Abstract. There is discrepancy between the occurrence of PDAC core mutations (Kras, Tp53, Cdkn2a, Smad4) reported here and in the Result section; please, amend. It is also confusing in the abstract when the authors initially state that the analysis has been conducted on 116 specimens but then report the results on 114 without explain the reasons. Finally, the conclusion of the abstract should be revised.
  • Cohort. The authors describe an initial cohort of 166 cases, of which 116 passed the quality control before sequencing analysis. Unfortunately, I cannot find detailed information about the criteria that were used to qualify/disqualify cases, e.g., quantity of available DNA, quality of the DNA, and/or neoplastic cell content. I think this is a relevant information and should be reported in the results section. Of the 116 cases underwent sequencing, mutations could be detected in 114. I think the authors should at least comment for the possible reasons that explain the lack of mutations in those two samples (few neoplastic nuclei for example).
  1. Clinical characteristics of the cohort. This is a very interesting pancreatic cancer cohort as it is composed almost exclusively by advanced cases. For this reason, I think it is extremely important a better description of the clinical characteristics of the cohort. First, the authors should make clear that they have clinical and not pathological staging information for the majority of cases given that only a small proportion of cancers were actually resected. Along this line, I think that more details should be given about the availability of data for all cases; I mean, was nodal status assessed for all cases? I found quite difficult to believe that the nodal status (N0, N1, N2?) was present for all cases just based on imaging analysis, and from the data provided I cannot understand whether Nx are actually present or not. The same applies to tumour mass/size information. Finally, 75 cases were metastatic and of those 30 were not liver metastases. However, I cannot find information on the specific metastatic sites (e.g., how many lungs, peritoneum or others).
  2. Clinical models. In general, the authors should provide a better description of the approach/methods used for building clinical models. I am not an expert and therefore I found difficult to follow the reasoning behind the inclusion of parameters, such as age, into the model given that it is a variable not associated to OS in the initial analysis (this specifically related to Figure 3). I found also quite odd that T and N status were not associated to survival, which is in sharp contrast with many previous reports (especially for the N status). Can this be due to the incomplete information about T and N? The same concern relates to the model described in Figure 6 with the regard to the inclusion of age as variable.
  3. Sequencing analysis. The authors used a cancer-only pipeline and therefore the description of the method used to call somatic mutations needs to be detailed. With cancer-only pipeline, there is a risk of reporting single-nucleotide polymorphisms and further not being able to assess biallelic inactivation for tumour suppressor genes. This is quite important, especially because the authors are reporting and exploring the significance of several tumour suppressor genes whose alterations can be also inherited in the germline (BRCA, NF1 and ATM). I would be very cautious in suggesting that BRCA2 mutations are associated to a metastatic phenotype and I would make sure of describing pathogenic mutations only.
  • It is completely obscure to me the reasoning behind the inclusion of rare events into the clinical models. How many NF1, FLT4, JAK1 mutations are detected in the entire cohort? Is the inclusion of those mutations just a proxy for a tumor mutational burden assessment? Please clarify.

Author Response

Dear Reviewer,

I appreciate your review and comments.

We sincerely did our best to respond to your comments.

Best regards, 

Joo Kyung Park

Reviewer 2 Report

The paper by Hyemin Kim et al. reports a cohort of 116 patients with pancreatic cancer undergoing NGS on EUS-acquired tissue. The study is of interest and would be worth being published, however I have the following suggestions to improve the quality of the paper:

  • the authors state that 64.6% of these patients were stage IV at diagnosis. In these cases guidelines suggest to perform sampling of the metastatic lesions. How come EUS was performed? this should be clarified
  • the authors state that 64.6% were stage IV, the most common site of metastases was the liver but then a percentage of 38.8% is mentioned and in the text it is not clear what this refers to (it is clear in the table though). This should be better explained
  • it is not clear in the methods section how the model was built and how they defined the cut off of high risk vs low risk pdac; this should be better explained
  • in table 2 no other mutation beside BRCA is reported? it would be interesting to see also for other mutations how the metastatic spread goes
  • in figure 6 in the table there should be also the 95% CI beside the raw HR, as some data are not clear (ca19.9 with HR=1 but a significant p)
  • in figure 6 in the table there are some p=0 that the authors might want to change to p<....
  • in kaplan-meier curves (at least the one in figure 6) it would be interesting to see the number of patients under the X line
  • the authors report that "Only, the validation of genetic alteration related to the survival in pancreatic cancer was performed with the TCGA database". This nevertheless does not seem to be mentioned in the methods section and should be specified
  • how the Clinical and Clinico-Genomic model was built should be more extensively specified
  • how the EUS-FNB samples were handled (formalin fixed paraffin embedded? from cytologic smear?) and DNA extracted should be better specified
  • the quantity of DNA extracted should be mentioned
  • the introduction and discussion should be expanded with some mentioning of other papers doing NGS panels on EUS samples and with some comparison to them being reported

Author Response

(The authors gave the same response as above.)

Reviewer 3 Report

In a retrospective study, the authors evaluated the feasibility of performing deep sequencing of minimal specimens from EUS-FNB, as well as using clinical factors and gene mutations to predict survival in PDAC. 116 samples from 166 EUS-FNB specimens passed the quality control test in order to obtain genomic profiles through deep sequencing. In addition, certain genetic alterations, such as single nucleotide variant of NF1, copy number variation of SMAD4, and loss of function mutation of CDKN2A were associated with survival. 

The authors describe in detail the process of obtaining EUS-FNB specimens and how targeted sequencing is performed.  

It would be helpful to discuss how the success rate of being able to perform deep sequencing from EUS-FNB samples compares with historical rates from surgical specimens. The authors discuss the clinical factors that were associated with overall survival using univariate analysis, but was there any multivariate analysis? In addition, it would be interesting to see if certain genetic alterations are associated with the significant clinical factors affecting prognosis, such as CA19-9, stage, or tumor location? Given that the study included patients with various stages of disease, different mutations with different treatments, it’s difficult to directly assess the impact of a clinic-genomic model in predicting survival. In addition, the clinic-genomic model has limited utility since it was based on patients who had EUS-FNB performed, which excludes patients who proceeded directly to surgical resection, who may have had better prognosis.  In addition it was only performed on the primary tumor and we know that metastatic lesions might have a different profile.   It would be beneficial for the authors to discuss how they came up with a validation model or how they used the model to successfully predict the survival in high-risk and low-risk groups.  

This is a novel study on the feasibility of performing targeted deep sequencing on EUS-FNB specimens, which there are limited studies on for PDAC. This may be of some utility given that the authors discuss how targeted deep sequencing can accurately predict prognosis. This can eventually lead to studies on optimal targeted therapy selection for PDAC. 

Author Response

(The authors gave the same response as above.)

Round 2

Reviewer 1 Report

The authors did a commendable job in addressing reviewers' concerns and I have no further questions/comments.

Reviewer 2 Report

The revised version of the manuscript significantly improved and all issues have been resolved. I therefore suggest publication in the present form.

Reviewer 3 Report

The authors provided a revision to their initial manuscript on the feasibility of performing deep sequencing of minimal specimens from EUS-FNB, as well as the clinical factors and gene mutations predictive of survival in PDAC. 

The authors address some of the concerns from the reviewers. The authors provide more detail about the clinical characteristics of PDAC cases, including common sites of metastatic disease, genetic mutations associated with metastatic disease, and that their study population may have had a worse prognosis than the excluded population that went to surgery directly. They also discuss that the frequencies of major mutations in PDAC were comparable to previous studies. By comparing their technical success rates with cytological, biopsy, and surgical tissue specimens, the authors strengthen the importance of their EUS-FNB specimens in deep sequencing. 

By including the additional revisions, the authors have provided additional value to the use of EUS-FNB for obtainment of deep sequencing specimens and its potential role in predicting disease prognosis of certain genetic alterations. However, the utility of determining genetic alterations is limited by the specific population studied in this retrospective review.